# FEDERATED LEARNING FOR LOCAL AND GLOBAL DATA DISTRIBUTION

**Gaurav Goswami**[1]**, Akshay Agarwal**[2]**, Nalini K. Ratha**[3]**, Richa Singh**[4]**, Mayank Vatsa**[4]
[1]IBM Singapore, [2]IISER Bhopal, India, [3]University at Buffalo, USA, [4]IIT Jodhpur, India
`gaurav.goswami@ibm.com, akagarwal@iiserb.ac.in, nratha@buffalo.edu`
`{richa, mvatsa}@iitj.ac.in`

## ABSTRACT

Existing research in Federated Learning focuses on synthetic or small-scale datasets, with in-house distribution posing challenges for long-term real-world use cases. We propose a novel approach that maximizes in-house (local) distribution gains while focusing on generalization. Experimental results on several datasets demonstrate the efficacy of the proposed approach.

## 1 INTRODUCTION

Federated Learning (FL) offers a solution to train machine learning models in a distributed manner without sacrificing data privacy. However, a major limitation of FL is that in cases where the data is not independently and identically distributed (also called non-IID) across clients, the global model does not perform well for clients' local datasets Li et al. (2019); Sattler et al. (2019); Shoham et al. (2019); Varno et al. (2022); Jhunjhunwala et al. (2022). To address this, in-house distribution or local adaptation techniques have been proposed which tune the global model to perform better on individual clients' local data Deng et al. (2020); T Dinh et al. (2020); Li et al. (2021); Ozkara et al. (2021); Hao et al. (2021); Huang et al. (2021); Sun et al. (2021); Ma et al. (2022); Tan et al. (2022). If this process is not regularized then we run the risk of creating biased models that excessively accommodate the client's distribution at the time of training and do not generalize well to the overall problem domain Abay et al. (2020); Zhang et al. (2020). This research proposes an approach for FL that balances in-house (local) distribution and generalization to mitigate this risk. The proposed Local and Global Federated Learning (LGFL) algorithm is evaluated on three databases.

## 2 LOCAL AND GLOBAL FEDERATED LEARNING

Current in-house distribution techniques create biased models by tuning them for the participant's data distribution that may sacrifice performance on unseen data. We propose that the success criteria of each local model should be two-fold: i) provide better performance on the local data than a model that could have been trained without FL, i.e., **in-house distribution** and ii) ensure that the model is not skewed and generalizes well to the problem domain, i.e., **generalization**. Existing local adaptation algorithms have primarily focused on addressing (i) but have not considered or evaluated (ii) which is the focus of the proposed approach. In-house distribution in federated learning can be defined as the performance of a model on a client's local data. We define generalization performance as the performance of the model on unseen test data from other clients. Due to the non-IID nature of the client-level data distribution, this ensures that the model is evaluated on a large dataset that will include data points from many different feature subspaces.

### 2.1 PROPOSED LGFL ALGORITHM

FL begins with an initial global model, $M_0^s$, sent to each client $C_i$, trained locally on the client's data $D_{train}^{C_i}$, and updated model weights are sent to the server. It aggregates these weights and creates the next iteration of the global model, $M_1^s$. This process is repeated $N$ times, and the final global model $M^s$ is created. Clients personalize $M^s$ and create a personalized model $M_i^c$ for each $C_i$.

In the proposed LGFL algorithm, we leverage a peer-review mechanism. After training a personalized model, $M_{Ci}$, each client $C_i$ sends the model to reviewing clients that provide a performance metric for the model on their data back to $C_i$ as their feedback score. Client $C_i$ can then stop the process if the metrics are satisfactory or continue the process and send the next iteration of the model

---

**Algorithm 1** LGFL: Proposed algorithm for local adaptation for a client $C_i$

---

Randomly select $N_{cr}$ clients as clients
**for** each round $j \in N$ **do**
    Perform local adaptation of client model $M_{Ci}$
    Evaluate in-house distribution performance $\rho(M_{Ci}, D_i)$ using $D_i$
    Evaluate generalization performance $\gamma(M_{Ci}, D_{cr})$ on $D_{cr}$ by sending $M_{Ci}$ to clients $N_{cr}$
**end for**
Select model trained in learning round $j$ with maximum value of $\alpha\rho(M_{Ci}, D_i) + \beta\gamma(M_{Ci}, D_{cr})$

---

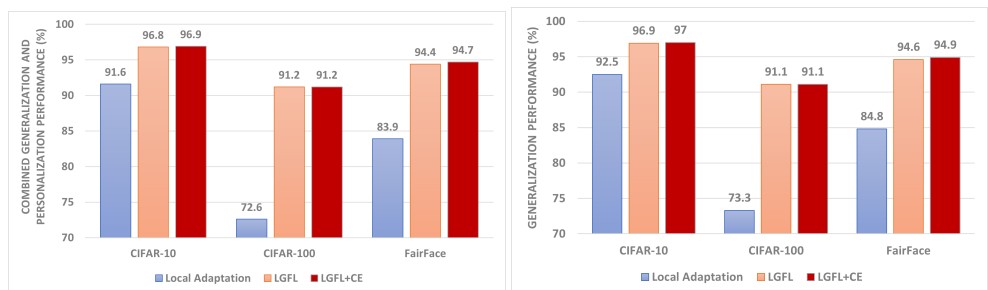

Figure 1: (left) Combined Local and Global evaluation results on the three databases. (right) Generalization performance is relative to the global model baseline, i.e., 100% generalization performance matches the global model. By using LGFL+CE, we are able to achieve up to **14.9%** best-case generalization performance gain over local adaptation (LA).

for review again. In this process, let $N_{cr}$ be the number of reviewers in a given cycle, then the updated optimization criteria for client $C_i$ becomes $\arg\max \alpha\rho(M_{Ci}, D_i) + \beta\gamma(M_{Ci}, D_{cr})$. Here, $\alpha$ and $\beta$ represent weights that can control the role of these factors in training with larger weights of $\beta$ representing more focus on preserving generalization performance while larger values of $\alpha$ allow more emphasis on improving in-house distribution performance.

Reviewers can be selected randomly ($N_{cr} = 10$) but to efficiently reduce the number of needed reviewers we propose client election ('**CE**') shown in Algorithm 2. The server can determine the similarity between clients' data by comparing client model weights using a layer-wise cosine similarity metric before the final round of training, allowing the client to select reviewers using this ranking and reducing communication costs by using fewer clients ($N_{cr} = 5$).

Clients do not share data with each other during the peer-review process, thereby maintaining data privacy and no additional information about client data is sent to the server as part of LGFL. While sharing the model can enable model inversion attacks, these are also a concern even for traditional FL. However, these can be addressed effectively with several defenses proposed in the literature Lewis et al. (2023); Ye et al. (2022).

## 3 EXPERIMENTAL ANALYSIS

We conduct experiments on three databases: CIFAR-10 Krizhevsky (2012), CIFAR-100 Krizhevsky (2012), and FairFace Karkkainen & Joo (2021). For all these databases, we first create non-IID splits such that the data distribution varies across clients in order to emulate a practical scenario.

Figure 1 presents the combined generalization and in-house distribution results on the three databases. The proposed LGFL+CE algorithm achieves up to 18.6% improved combined performance. The key takeaway is that the proposed LGFL algorithm provides guardrails to the applied in-house distribution algorithm (Local Adaptation in this case) and ensures that improvement in in-house distribution does not come at the cost of sacrificing the generalizability of the model. It functions with any FL/in-house distribution algorithm and therefore achieving higher in-house distribution performance is possible by utilizing better core models.

## 4 CONCLUSION

We proposed a novel approach for FL that allows leveraging gains from in-house distribution while keeping generalization in focus. Our proposed LGFL algorithm showed promising results on three databases and offers exciting research opportunities for the community.

URM STATEMENT

The authors acknowledge that the first author of this work meets the URM criteria of the ICLR 2023 Tiny Papers Track.

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

## A APPENDIX

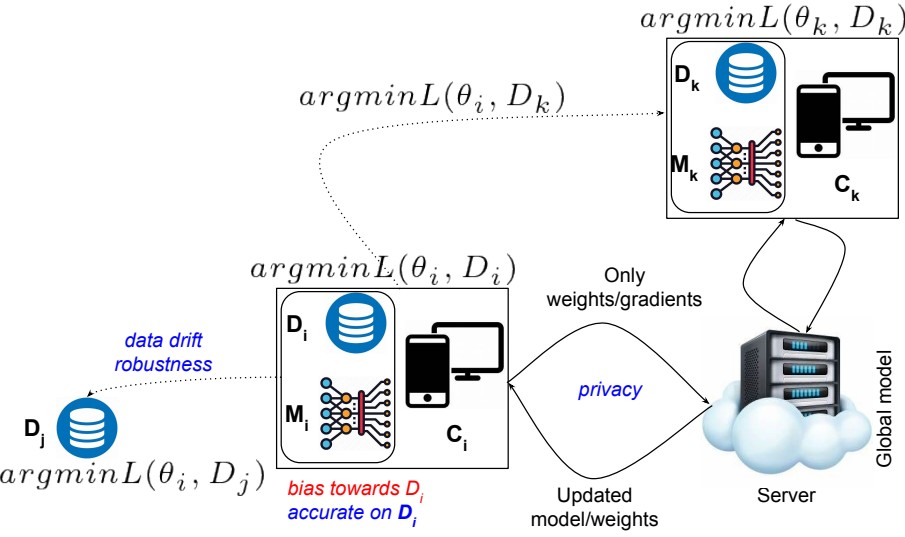

Figure 2: LGFL: Local and Global Federated Learning. $C_i$ denotes the $i^{th}$ agent in the FL system, $D_i$ denotes its local data, and $M_i$ its classifier with parameters $\theta_i$. A client receives a global model from the server and performs in-house distribution using its local data, improving the model's accuracy. However, in-house distribution in isolation can lead to an increase in the bias of the model. We introduce '*generalization*' into the existing approach which adds a second objective to the process by validating against other clients' data ($\arg\min L(\theta_i, D_k)$) to ensure robustness against scenarios that cause the data distribution to change over time.

---

**Algorithm 2** RS: Proposed reviewer selection algorithm

---

Let server have weight updates $\delta_i$ sent by each client $C_i$
**for** each participating client $C_i$ **do**
  **for** each participating client $C_j$ **do**
    **if** $i \neq j$ **then**
      Compute similarity metric $S_{ij} = \phi(\delta_i, \delta_j)$ where $\phi$ is a similarity metric function
    **end if**
  **end for**
  Sort clients based on ascending $S_{ij}, \forall j \neq i$
  Select top $N'_{cr}$ clients based on sorted $S_{ij}$ and reviewer agreement such that $N'_{cr} < N_{cr}$
**end for**

---

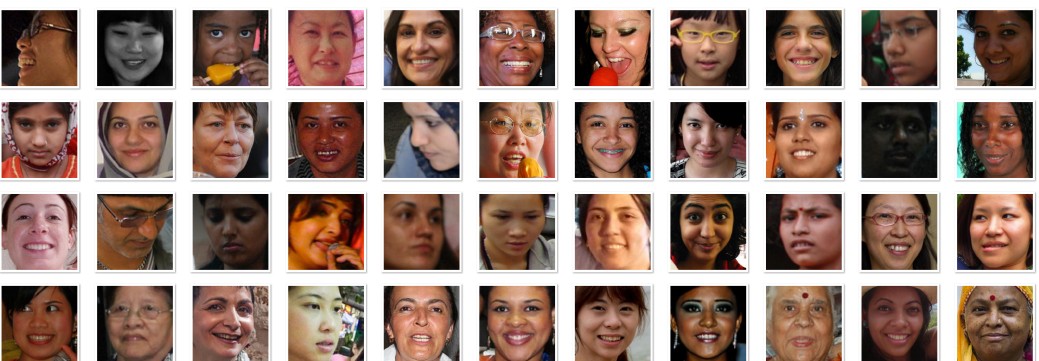

Figure 3: FairFace test images misclassified by an adapted client model personalized using LA that are correctly classified by the model adapted using the proposed LA+LGFL+RS algorithm. We can see that a large majority of these are from under-represented groups and the proposed algorithm can help avoid creating biased models by preserving generalizability.

