# OpenReview forum: "Federated Learning for Local and Global Data Distribution"
_ICLR.cc/2023/TinyPapers — Submitted to Tiny Papers @ ICLR 2023_

### Official Review · Reviewer_4iAv · 2023-03-22

**Confidence:** 4

**Summary Of Contributions:**

The article considers the impact of Federated Learning on the individual client's performance for their own local data (in-house distribution) and general unseen but similar data (generalization). The authors propose a solution that can optimize for both in-house and general distributions unlike prior methods.

**Rating:**

High Potential (HP): a submission which meets the reviewing criteria and has potential to make an impact on the field

**Strengths And Weaknesses:**

Strengths:
- The article investigates an interesting problem of improving both the in-house local distribution and generalization performance of local clients in a Federated Learning (FL) Scenario. The authors suggest that after FL convergence, each client can locally train models (local adaptation) and then submit its model to nearby clients for improving their generalization performance. By optimizing over a weighted generalization and personalization loss, the clients are better prepared for both kinds of problems.
- The experiments show significant improvement in the clients' generalization performance when they are trained with the proposed Local and Global Federated Learning (LFGL) algorithm rather than the basic local adaptation algorithm.

Weaknesses:

- Experiments comparing other baselines in this space can be useful to assess the importance of these results.
- Privacy and communication costs of LGFL due to model sharing over clients are not studied. Intuitively, both of these might be on the higher end.
- Results that show the possible decrease in personalization performance due to LGFL are missing.



**Suggested Changes:**

- For the Local and Global Federated Learning (LFGL) algorithm, it is unclear how the peer-review process proceeds when the clients are not selected by the Client Election (CE) algorithm. Furthermore, the privacy implications of directly sharing models trained on local data are unclear and unstudied.
- From the results, it looks like the LGFL algorithm primarily contributes to the strong combined generalization and personalization gains. Therefore, it is unclear if the CE algorithm which is also costly and scales quadratically with the number of clients is useful. Similar to LGFL, the privacy implications of finding similarity metrics for clients are unstudied. Adding these extra details can improve the contribution majorly.
- Theoretical basis and intuition for cross-sharing of models should be provided. For example, the number of peer-reviewing clients or rounds of training required beyond local adaptation for convergence needs to be studied.
- A minor comment for readability: Since the LGFL algorithm alone shows higher generalization gains, it might be useful to shift this algorithm in the paper's main section. Shifting the Client Election algorithm to the appendix should be okay. Furthermore, the article writing and definitions can be considerably improved.

---

> ### Author Response · Authors · 2023-05-25
> **Responding to key concerns**
>
> We thank the reviewer for the encouragement and the thoughtful and detailed feedback. We would like to try to address some of the key concerns that have been pointed out:
>
> **Concern:** *Privacy and communication costs of LGFL due to model sharing over clients are not studied. Intuitively, both of these might be on the higher end.*
>
> **Response:** Since the peer-review process does not involve any movement of the data between clients, data privacy is not compromised during the peer-review process. While model inversion attacks can attempt to reproduce training data using the model, this risk also exists in traditional federated learning when clients send model information to the server. Several defenses against such model inversion attacks have been proposed which can effectively mitigate these and the same can be leveraged in the proposed approach as well. Communication cost as compared to traditional FL is relatively higher, however, model weights can be effectively compressed before moving them. We also believe that the increase in cost may be acceptable for several use cases where the added communication cost is outweighed by the gain in generalizable performance of the resulting models.
>
> **Concern:** *Results that show the possible decrease in personalization performance due to LGFL are missing.*
>
> **Response:** Figure 1 (left) showcases the combined performance metrics which includes performance on the local data. We find that the net effect of personalization performance loss is not substantial when compared to the improvement in generalization performance. For use cases or clients where generalization is less of a concern, opting out or adjusting the weights of generalization and personalization performance (parameters alpha and beta) is a way to further control any loss of personalization depending on the specific characteristics of their data distribution and model behavior

---

### Official Review · Reviewer_U5W3 · 2023-03-30

**Confidence:** 5

**Summary Of Contributions:**

This paper proposes a peer-review mechanism for federated learning, in which multiple reviewing clients are selected for each client. During the training process, each client sends its local model to the reviewers and receives scores. At the end of the training, each client chooses the local model with the highest score as its final local model.

**Rating:**

Needs Clarification (NC): a submission which does not meet the reviewing criteria and needs clarification for its described problem or solution

**Strengths And Weaknesses:**

**Strengths**
- The paper proposes an interesting algorithm for federated learning.
- The submission of this paper meets the formatting requirements and page limits.

**Weaknesses**
- The paper claims that existing methods only focus on the performance of in-house distribution and neglect the generalization ability of the model for the overall problem. However, this claim is inaccurate, as most of the non-iid methods in the literature report their accuracy on a global test set.
- The description of the method is too vague, and only the selection criterion for the review clients is given in the main text.
- The proposed method simply selects the best-performing model among those obtained from multiple rounds as the final local model. I do not think this method has any value for publication.
- The algorithm description in the appendix seems to be missing a key part, which is the interaction process with the central server. Is this a decentralized framework?
- The paper points out that current research only focuses on small or simulated datasets, yet the experiments in this paper are also conducted on small datasets.
- The paper does not provide any experimental results for other non-iid methods.

**Suggested Changes:**

- The authors need to provide a detailed description of their method in the main text. Currently, the main text only describes how they selected the review clients.
- They need to add comparative experiments with other non-iid methods. There are no experimental results from any relevant non-iid papers for comparison.

---

> ### Author Response · Authors · 2023-05-25
> **Responding to key concerns**
>
> We thank the reviewer for the thoughtful and detailed feedback. We would like to try to address some of the key concerns that have been pointed out:
>
> **Concern:** *The paper claims that existing methods only focus on the performance of in-house distribution and neglect the generalization ability of the model for the overall problem. However, this claim is inaccurate, as most of the non-iid methods in the literature report their accuracy on a global test set.*
>
> **Response:** While general non-iid addressing federated learning contributions do evaluate performance on a global test set, here we are referring primarily to many proposed personalization techniques that typically evaluate performance only on local client data and not a global set.
>
> **Concern:** *The proposed method simply selects the best-performing model among those obtained from multiple rounds as the final local model. I do not think this method has any value for publication.*
>
> **Response:** The proposed method's novelty lies in evaluating the model's performance using the peer review mechanism in the absence of large test data available to each client without needing the sharing of data amongst clients or from clients to the server.
>
> **Concern:** *The algorithm description in the appendix seems to be missing a key part, which is the interaction process with the central server. Is this a decentralized framework?*
>
> **Response:** It is indeed a decentralized framework during the personalization process and the server's role ends after suggesting the list of reviewers to each client. After that, the clients communicate the scores directly with each other without requiring server intervention.

---

### Author Response · Authors · 2023-05-25
**Summary of revisions made**

We thank the reviewers and the area chair again for their kind suggestions to improve the paper. We have made a few revisions to account for these suggestions and would like to present a summary of the key revisions we have incorporated:

1. We have moved the LGFL algorithm to the main text and CE to the appendix
2. We have clarified the client selection approach in absence of CE (random) and also noted the impact on the number of needed reviewers by specifying the exact number of reviewers used in random selection vs proposed CE (10 for random selection which is the base LGFL result, 5 for CE). We are effectively able to halve the reviewers needed while improving results by using the proposed CE algorithm. This also helps highlight the contribution of the CE algorithm.
3. We have added a note to address data privacy concerns (in the proposed algorithm section) that clarifies that no data is shared between clients during peer review and that no additional data or information about client datasets is sent by the clients to the server.

---

### Comment · Area_Chair_jKhg · 2023-06-02
**Archival**

This work meets the threshold for archival, contents the URM statement and is deanonymized

---

### Meta-Review · Area_Chair_jKhg · 2023-04-04

**Recommendation:** Invite to archive
**Confidence:** 3

**Metareview:**

The paper proposes a federated learning approach that preserves generalization during personalization and experimental results are presented to show the efficacy of the methodology.

**Summary:**

The reviews are mixed but some of the weaknesses can be addressed by the authors.

**Comments And Feedback To The Authors:**

Please consider some of the suggestions mentioned by the reviewers. This paper has potential to be presented at the conference should those concerns be addressed.

**Reason For Not Giving A Higher Recommendation:**

As one reviewer points out, the claim that existing methods only focus on the performance of in-house distribution may not be true.

**Reason For Not Giving A Lower Recommendation:**

Some of the other weaknesses mentioned by the reviewers like lack of results on large datasets etc may not be applicable for the scope of this paper/venue. One of the main concerns expressed by the reviewers is that of privacy while calculating similarities between client data. This is certainly valid and could be a good future research starting point.

---

> ### Author Response · Authors · 2023-05-25
> **Responding to key concerns**
>
> We thank the area chair for the consideration and the thoughtful feedback. We would like to address the two key concerns:
>
> **Concern:** *As one reviewer points out, the claim that existing methods only focus on the performance of in-house distribution may not be true.*
>
> **Response:** We have found that while general federated learning contributions that focus on non-iid scenarios do perform evaluations on a global test set, many proposed personalization techniques typically evaluate and focus on performance only on local client data and not a global set. With this paper, we wish to emphasize on the importance of generalization in this process especially for many use cases where this may cause an unforeseen trade-off leading to suboptimal efficacy of personalized models. The proposed approach makes the trade-off more explicit and controllable while providing a method by which to do this effectively while preserving data privacy.
>
> **Concern:** *One of the main concerns expressed by the reviewers is that of privacy while calculating similarities between client data. This is certainly valid and could be a good future research starting point.*
>
> **Response:** The server uses only the model weights provided by each client to indirectly estimate similarities between client data based on the proximity of the inner layers of the respective models. No client data or client data characteristics are sent to the server. The data sent by the clients to the server, i.e. model weights or weight deltas, is the same as in the typical federated learning flow and therefore, we believe that the proposed approach does not introduce additional data privacy concerns with this similarity computation.

---

### Decision · Program_Chairs · 2023-04-10

Invite to archive

---

> ### Author Response · Authors · 2023-05-30
> **Archive**
>
> We wish to opt-in for archival (publishing the paper).
>
> Thanks